# THEORETICAL PROPERTIES OF THE GLOBAL OPTIMIZER OF TWO-LAYER NEURAL NETWORK

## ABSTRACT

In this paper, we study the problem of optimizing a two-layer artificial neural network that best fits a training dataset. We look at this problem in the setting where the number of parameters is greater than the number of sampled points. We show that for a wide class of differentiable activation functions (this class involves most nonlinear functions and excludes piecewise linear functions), we have that arbitrary first-order optimal solutions satisfy global optimality provided the hidden layer is non-singular. We essentially show that these non-singular hidden layer matrix satisfy a "good" property for these big class of activation functions. Techniques involved in proving this result inspire us to look at a new algorithmic, where in between two gradient step of hidden layer, we add a stochastic gradient descent (SGD) step of the output layer. In this new algorithmic framework, we extend our earlier result and show that for all finite iterations the hidden layer satisfies the "good" property mentioned earlier therefore partially explaining success of noisy gradient methods and addressing the issue of data independency of our earlier result. Both of these results are easily extended to hidden layers given by a flat matrix from that of a square matrix. Results are applicable even if network has more than one hidden layer provided all inner hidden layers are arbitrary, satisfy non-singularity, all activations are from the given class of differentiable functions and optimization is only with respect to the outermost hidden layer. Separately, we also study the smoothness properties of the objective function and show that it is actually Lipschitz smooth, i.e., its gradients do not change sharply. We use smoothness properties to guarantee asymptotic convergence of $O(1/\text{number of iterations})$ to a first-order optimal solution.

## 1 INTRODUCTION

Neural networks architecture has recently emerged as a powerful tool for a wide variety of applications. In fact, they have led to breakthrough performance in many problems such as visual object classification (Krizhevsky et al., 2012), natural language processing (Collobert & Weston, 2008) and speech recognition (Mohamed et al., 2012). Despite the wide variety of applications using neural networks with empirical success, mathematical understanding behind these methods remains a puzzle. Even though there is good understanding of the representation power of neural networks (Barron, 1994), training these networks is hard. In fact, training neural networks was shown to be NP-complete for single hidden layer, two node and $\text{sgn}(\cdot)$ activation function (Blum & Rivest, 1988). The main bottleneck in the optimization problem comes from non-convexity of the problem. Hence it is not clear how to train them to global optimality with provable guarantees.

Neural networks have been around for decades now. A sudden resurgence in the use of these methods is because of the following: Despite the worst case result by Blum & Rivest (1988), first-order methods such as gradient descent and stochastic gradient descent have been surprisingly successful in training these networks to global optimality. For example, Zhang et al. (2016) empirically showed that sufficiently over-parametrized networks can be trained to global optimality with stochastic gradient descent.

Neural networks with zero hidden layers are relatively well understood in theory. In fact, several authors have shown that for such neural networks with monotone activations, gradient based methods will converge to the global optimum for different assumptions and settings (Mei et al., 2017; Hazan et al., 2015; Kakade et al., 2011; Kalai & Sastry, 2009).

Despite the hardness of training the single hidden layer (or two-layer) problem, enough literature is available which tries to reduce the hardness by making different assumptions. E.g., Choromanska et al. (2014) made a few assumptions to show that every local minimum of the simplified objective is close to the global minimum. They also require some independent activations assumption which may not be satisfied in practice. For the same shallow networks with (leaky) ReLU activations, it was shown in Soudry & Carmon (2016) that all local minimum are global minimum of the modified loss function, instead of the original objective function. Under the same setting, Xie et al. (2016) showed that critical points with large "diversity" are near global optimal. But ensuring such conditions algorithmically is difficult.

All the theoretical studies have been largely focussed on ReLU activation but other activations have been mostly ignored. In our understanding, this is the first time a theoretical result will be presented which shows that for almost all nonlinear activation functions including softplus, an arbitrary first-order optimal solution is also the global optimal provided certain "simple" properties of hidden layer. Moreover, we show that a stochastic gradient descent type algorithm will give us those required properties for free for all finite number of iterations hence even if the hidden layer variables are data dependent, we still get required properties. Our assumption on data distribution is very general and can be reasonable for practitioners. This comes at two costs: First is that the hidden layer of our network can not be wider than the dimension of the input data, say $d$. Since we also look at this problem in over-parametrized setting (where there is hope to achieve global optimality), this constraint on width puts a direct upper-bound of $d^2$ on the number of data points that can be trained. Even though this is a strong upper bound, recent results from margin bounds (Neyshabur et al., 2017) show that if optimal network is closer to origin then we can get an upper bound on number of samples independent of dimension of the problem which will ensure closeness of population objective and training objective. Second drawback of this general setting is that we can prove good properties of the optimization variables (hidden layer weights) for only finite iterations of the SGD type algorithm. But as it is commonly known, stochastic gradient descent converges to first order point asymptotically so ideally we would like to prove these properties for infinitely many iterations. We compare our results to some of the prior work of Xie et al. (2016) and Soudry & Carmon (2016). Both of these papers use similar ideas to examine first order conditions but give quite different results from ours. They give results for ReLU or Leaky ReLU activations. We, on the other hand, give results for most other nonlinear activations, which can be more challenging. We discuss this in section 3 in more detail.

We also formally show that even though the objective function for training neural networks is non-convex, it is Lipschitz smooth meaning that gradient of the objective function does not change a lot with small changes in underlying variable. To the best of our knowledge, there is no such result formally stated in the literature. Soltanolkotabi et al. (2017) discuss similar results, but there constant itself depends locally on $w_{max}$, a hidden layer matrix element, which is variable of the the optimization function. Moreover, there result is probabilistic. Our result is deterministic, global and computable. This allows us to show convergence results for the gradient descent algorithm, enabling us to establish an upper bound on the number of iterations for finding an $\varepsilon$-approximate first-order optimal solution ($\|\nabla f()\| \leq \varepsilon$). Therefore our algorithm will generate an $\varepsilon$-approximate first-order optimal solution which satisfies aforementioned properties of the hidden layer. Note that this does not mean that the algorithm will reach the global optimal point asymptotically. As mentioned before, when number of iterations tend to infinity, we could not establish "good" properties. We discuss technical difficulties to prove such a conjecture in more detail in section 5 which details our convergence results.

At this point we would also like to point that there is good amount of work happening on shallow neural networks. In this literature, we see variety of modelling assumptions, different objective functions and local convergence results. Li & Yuan (2017) focuses on a class of neural networks which have special structure called "Identity mapping". They show that if the input follows from Gaussian distribution then SGD will converge to global optimal for population objective of the "identity mapping" network. Brutzkus & Globerson (2017) show that for isotropic Gaussian inputs, with one hidden layer ReLU network and single non-overlapping convolutional filter, all local minimizers are global hence gradient descent will reach global optimal in polynomial time for the population objective. For the same problem, after relaxing the constraint of isotropic Gaussian inputs, they show that the problem is NP-complete via reduction from a variant of set splitting problem. In both of these studies, the objective function is a population objective which is significantly different from training objective in over parametrized domain. In over-parametrized regime, Soltanolkotabi et al. (2017) shows that for the training objective with data coming from isotropic Gaussian distribution,

provided that we start close to the true solution and know maximum singular value of optimal hidden layer then corresponding gradient descent will converge to the optimal solution. This is one of its kind of result where local convergence properties of the neural network training objective function have studied in great detail.

Our result differ from available current literature in variety of ways. First of all, we study the training problem in the over-parametrized regime. In that regime, the training objective can be significantly different from population objective. Moreover, we study the optimization problem for many general non-linear activation functions. Our result can be extended to deeper networks when considering the optimization problem with respect to outermost hidden layer. We also prove that stochastic noise helps in keeping the aforementioned properties of hidden layer. This result, in essence, provides justification for using stochastic gradient descent.

Another line of study looks at the effect of over-parametrization in the training of neural networks (Haeffele & Vidal, 2015; Nguyen & Hein, 2017). These result are not for the same problem as they require huge amount of over-parametrization. In essence, they require the width of the hidden layer to be greater than number of data points which is unreasonable in many settings. These result work for fairly general activations as do our results but we require a moderate over-parametrization, width $\times$ dimension $\geq$ number of data population, much more reasonable in practice as pointed before from margin bound results. They also work for deeper neural network as do our results when optimization is with respect to outermost hidden layer (and aforementioned technical properties are satisfied for all hidden layers).

## 2 NOTATION AND PROBLEM OF INTEREST

We define set $[q] := \{1, \ldots, q\}$. For any matrix $A \in \mathbb{R}^{a \times b}$, we write $\text{vect}(A) \in \mathbb{R}^{ab \times 1}$ as vector form of the matrix $A$. For any vector $z \in \mathbb{R}^k$, we denote $h(z) := \begin{bmatrix} h(z[1]), \ldots, h(z[k]) \end{bmatrix}^T$, where $z[i]$ is the $i$-th element in vector $z$. $\mathcal{B}^i(r)$ represents a $l_i$-ball of radius $r$, centred at origin. We define component-wise product of two vectors with operator $\odot$.

We say that a collection of vectors, $\{v^i\}_{i=1}^N \in \mathbb{R}^d$, is full rank if $\text{rank}\Big( \begin{bmatrix} v^1 & \ldots & v^N \end{bmatrix} \Big) = \min\{d, N\}$. Similarly, we say that collection of matrices, $\{M_i\}_{i=1}^N \in \mathbb{R}^{n \times d}$, is full rank if $\text{rank}\Big( \begin{bmatrix} \text{vect}(M_1) & \ldots & \text{vect}(M_k) \end{bmatrix} \Big) = \min\{N, nd\}$.

A fully connected two-layer neural network has three parameters: hidden layer $W$, output layer $\theta$ and activation function $h$. For a given activation function, $h$, we define neural network function as

$$\phi_{W,\theta}(u) := \theta^T h(Wu).$$

In the above equation, $W \in \mathbb{R}^{n \times d}$ is hidden layer matrix, $\theta \in \mathbb{R}^n$ is the output layer. Finally $h : \mathbb{R} \to \mathbb{R}$ is an activation function.

The main problem of interest in this paper is the two-layer neural network problem given by

$$\min_{\substack{W \in \mathbb{R}^{n \times d} \\ \theta \in \mathbb{R}^n}} f(W, \theta) := \frac{1}{2N} \sum_{i=1}^N (v^i - \phi_{W,\theta}(u^i))^2. \tag{2.1}$$

In this paper, we assume that $(u^i, v^i) \in \mathbb{R}^d \times \mathbb{R}, i \in [N]$ are independently distributed data point and each $u^i$ is sampled from a $d$-dimensional Lebesgue measure.

## 3 THE BASIC IDEA AND THE ALGORITHM

First-order optimality condition for the problem defined in (2.1), with respect to $W[j, k]$ (j-th row, k-th column element of matrix $W$) $\forall j \in [n], \forall k \in [d]$ is

$$\nabla_W f(W, \theta)[j, k] = \frac{1}{N} \sum_{i=1}^N \{v^i - \theta^T h(Wu^i)\} h'(W[j, :]u^i) \theta[j] u^i[k] = 0. \tag{3.1}$$

Equation (3.1) is equivalent to

$$\sum_{i=1}^N \{v^i - \theta^T h(Wu^i)\} \big( h'(Wu^i) \odot \theta \big) u^{i^T} = \mathbf{0}. \tag{3.2}$$

(3.1) can also be written in a matrix vector product form:

$$Ds = \mathbf{0}, \tag{3.3}$$

where

$$D := \begin{bmatrix} h'(W[1,:]u^1)\theta[1]u^1 & \dots & h'(W[1,:]u^N)\theta[1]u^N \\ \vdots & \ddots & \vdots \\ h'(W[d,:]u^1)\theta[d]u^1 & \dots & h'(W[d,:]u^N)\theta[d]u^N \end{bmatrix} \text{ and } s := \begin{bmatrix} v^1 - \theta^T h(Wu^1) \\ \vdots \\ v^N - \theta^T h(Wu^N) \end{bmatrix}.$$

Notice that if matrix $D \in \mathbb{R}^{nd \times N}$ is of full column rank (which implies $nd \geq N$, i.e., number of samples is less than number of parameters) then it immediately gives us that $s = 0$ which means such a stationary point is global optimal. This motivates us to investigate properties of $h$ under which we can provably keep matrix $D$ full column rank and develop algorithmic methods to help maintain such properties of matrix $D$.

Note that similar approach was explored in the works of Soudry & Carmon (2016) and Xie et al. (2016). To get the full rank property for matrix $D$, Soudry & Carmon (2016) use leaky ReLu function. Basically this leaky activation function adds noise to entries of matrix $D$ which allows them to show matrix $D$ is full rank and hence all local minimums are global minimums. So this is essentially a change of model. We, on the other hand, do not change model of the problem. Moreover, we look at the algorithmic process of finding $W$ differently. We show that SGD will achieve full rank property of matrix $D$ with probability 1 for all finite iterations. So this is essentially a property of the algorithm and not of the model. Even if that is the case, to show global optimality, we need to prove that matrix $D$ is full column rank in asymptotic sense.

That question was partly answered in Xie et al. (2016). They show that matrix $D$ is full column rank by achieving a lower bound on smallest singular value of matrix $D$. But to get this, they need two facts. First, the activation function has to be ReLu so that they can find the spectrum of corresponding kernel matrix. Second, they require a bound on discrepancy of weights $W$. These conditions are strong in the sense that they restrict the analysis to a non-differentiable activation function and finding an algorithm satisfying discrepancy constraint on $W$ can be a difficult task. On other hand, our results are proved for a simple SGD type algorithm which is easy to implement. But we do not get a lower bound on singular value of $D$ in asymptotic sense. There are obvious pluses and minuses for both types of results.

For the rest of the discussion, we will assume that $n = d$ (our results can be extended to case $n \leq d$ easily) and hence $W$ is a square matrix. In this setting, we develop the following algorithm whose output is a provable first-order approximate solution. Here we present the algorithm and in next sections we will discuss conditions that are required to satisfy full rank property of matrix $D$ as well as convergence properties of the algorithm.

In Algorithm 1, we use techniques inspired from alternating minimization to minimize with respect to $\theta$ and $W$. For minimization with respect to $\theta$, we add gaussian noise to the gradient information. This will be useful to prove convergence of this algorithm. We use randomness in $\theta$ to ensure some "nice" properties of $W$ which help us in proving that matrix $D$ generated along the trajectory of Algorithm 1 is full column rank. More details will follow in next section.

The algorithm has two loops. An outer loop implements a single gradient step with respect to hidden layer, $W$. For each outer loop iteration, there is an inner loop which optimizes objective function with respect to $\theta$ using a stochastic gradient descent algorithm. In the stochastic gradient descent, we generate a noisy estimated of $\nabla_\theta f(W, \theta)$ as explained below.

Let $\xi \in \mathbb{R}^d$ be a vector whose elements are i.i.d. Gaussian random variable with zero mean. Then for a given value of $W$ we define stochastic gradient w.r.t. $\theta$ as follows:

$$G_W(\theta, \xi) = \nabla_\theta f(W, \theta) + \xi. \tag{3.4}$$

Then we know that

$$\mathbb{E}[G_W(\theta, \xi)] = \nabla_\theta f(W, \theta).$$

We can choose a constant $\sigma > 0$ such that following holds

$$\mathbb{E}\left[\left\|G_W(\theta, \xi) - \nabla_\theta f(W, \theta)\right\|^2\right] \leq \sigma^2. \tag{3.5}$$

Moreover, in the algorithm we consider a case where $\theta \in \mathcal{R}$. Note that $\mathcal{R}$ can be kept equal to $\mathbb{R}^d$ but that will make parameter selection complicated. In our convergence analysis, we will use

$$\mathcal{R} := \mathcal{B}^2(R/2), \tag{3.6}$$

for some constant $R$, to make parameter selection simpler. We use prox-mapping $P_x : \mathbb{R}^d \to \mathcal{R}$ as follows:

$$P_x(y) = \operatorname*{argmin}_{z \in \mathcal{R}} \langle y, z - x \rangle + \frac{1}{2}\|z - x\|^2. \tag{3.7}$$

In case $\mathcal{R}$ is a ball centred at origin, solution of (3.7) is just projection of $x - y$ on that ball. For case where $\mathcal{R} = \mathbb{R}^d$ then the solution is quantity $x - y$ itself.

---

**Algorithm 1** SGD-GD Algorithm

---

**procedure**
    $W_0 \leftarrow$ Random $d \times d$ matrix
    $\theta_0 \leftarrow$ Random $d$ vector
    Initialize $N_o$ to predefined iteration count for outer ietaration
    Initialize $N_i$ to predefined iteration count for inner iteration
    Begin **outer iteration**:
    **for** $k = 0, 1, 2, \ldots, N_o$ **do**
        $\bar{\theta}_1 \leftarrow \theta_k$
        Begin **inner iteration**:
        **for** $i = 1, 2, \ldots, N_i$ **do**
            $\bar{\theta}_{i+1} \leftarrow P_{\bar{\theta}_i}(\beta_i G_{W_k}(\bar{\theta}_i, \xi_i^k))$
            $\bar{\theta}_{i+1}^{av} = \left( \sum_{\tau=1}^{i} \beta_\tau \right)^{-1} \sum_{\tau=1}^{i} \beta_\tau \bar{\theta}_{\tau+1}$
        **end for**
        $\theta_{k+1} \leftarrow \bar{\theta}_{N_i+1}^{av}$
        $W_{k+1} \leftarrow W_k - \gamma_k \nabla_W f(W_k, \theta_{k+1})$
    **end for**
    **return** $\{W_{N_o+1};\ \theta_{N_o+1}\}$
**end procedure**

---

Notice that the problem of minimization with respect to $\theta$ is a convex minimization problem. So we can implement many procedures developed in the Stochastic optimization literature to get the convergence to optimal value (Nemirovski et al., 2009).

In the analysis, we note that one does not even need to implement complete inner iteration as we can skip the stochastic gradient descent suboptimally given that we improve the objective value with respect to where we started, i.e.,

$$f(W_k, \theta_{k+1}) \leq f(W_k, \theta_k). \tag{3.8}$$

In essence, if evaluation of $f$ for every iteration is not costly then one might break out of inner iterations before running $N_i$ iterations. If it is costly to evaluate function values then we can implement the whole SGD for convex problem with respect to $\theta$ as specified in inner iteration of the algorithm above. In each outer iteration, we take one gradient decent step with respect to variable $W$. We have total of $N_o$ outer iterations. So essentially we evaluate $\nabla_\theta f(W, \cdot)$ a total of $N_o N_i$ times and $\nabla_W f(\cdot, \theta)$ total of $N_o$ times.

Overall, this algorithm is new form of alternate minimization, where one iteration can be potentially left suboptimally and other one is only one gradient step.

## 4   ARBITRARY POINT SATISFYING FIRST ORDER OPTIMALITY CONDITIONS

We prove in this section that arbitrary first order optimal points are globally optimal. One does not expect to have an arbitrary first order optimal point because it has to depend on data. We still would like to put our analysis here because that inspires us to consider a new algorithmic framework in Section 3 providing similar results for all finite iterations of the algorithm.

We say that $h : \mathbb{R} \to \mathbb{R}$ satisfy the condition "**C1**" if

- $\forall$ interval $(a, b)$, $\nexists \{c_1, c_2, c_3\} \in \mathbb{R}^3$ s.t.

$$\{h'(x) = c_1, \forall x \in (a, b)\} \ or$$
$$\{(x + c_2)h'(x) + h(x) = c_3, \forall x \in (a, b)\}.$$

One can easily notice that most activation functions used in practice e.g.,

- *(Softplus)* $h(x) := ln(1 + e^x)$,
- *(Sigmoid)* $h(x) := \frac{1}{1+e^{-x}}$,
- *(Sigmoid symmetric)* $h(x) := \frac{1-e^{-x}}{1+e^{-x}}$,
- *(Gaussian)* $h(x) := e^{-x^2}$,
- *(Gaussian Symmetric)* $h(x) := 2e^{-x^2} - 1$,
- *(Elliot)* $h(x) := \frac{x}{2(1+|x|)} + 0.5$,
- *(Elliot Symmetric)* $h(x) := \frac{x}{1+|x|}$,
- *(Erf)* $h(x) := \frac{2}{\sqrt{\pi}} \int_0^x e^{-t^2/2} dt$,
- *(Hyperbolic tangent)* $h(x) := \tanh(x)$,

satisfy the condition **C1**. Note that $h'(x)$ also satisfy condition **C1** for all of them. In fact, except for very small class of functions (which includes linear functions), none of the continuously differentiable functions satisfy condition **C1**.

We first prove a lemma which establishes that columns of the matrix $D$ (each column is a vector form of $d \times d$ matrix itself) are linearly independent when $W = I_d$ and $h'$ satisfies condition **C1**. We later generalise it to any full rank $W$ using a simple corollary. The statement of following lemma is intuitive but its proof is technical.

**Lemma 4.1** *Suppose $x^i \in \mathbb{R}^d$ are independently chosen vectors from any d-dimensional Lebesgue measure and let $h : \mathbb{R} \to \mathbb{R}$ be any function that satisfies condition **C1** then collection of matrices $h(x^i)x^{i^T}, i \in [N]$ are full rank with measure 1.*

Now Lemma 4.1 gives us a simple corollary:

**Corollary 4.2** *If W is a nonsingular square matrix and $u^i \in \mathbb{R}^d$ is independently sampled from a Lebesgue measure then the collection of matrices $\left\{ h(Wu^i)u^{i^T} \right\}_{i=1}^N$ is full rank with measure 1.*

This means that if $u^i$ in the Problem (2.1) are coming from a Lebesgue measure then by corollary 4.2 we have $h(Wu^i)u^{i^T}$ will be a full rank collection given that we have maintained full rank property of $W$. Now note that in the first-order condition, given in (3.3), row of matrix $D$ are scaled by constant factors $\theta[j]$'s, $j \in [d]$. Notice that we may assume $\theta[j] \neq 0$ because otherwise there is no contribution of corresponding $j$-th row of $W$ to the Problem (2.1) and we might as well drop it entirely from the optimization problem. Hence we can rescale rows of matrix $D$ by factor $\frac{1}{\theta[j]}$ without changing the rank. In essence, corollary 4.2 implies that matrix $D$ is full rank when $W$ is full rank. So by our discussion in earlier section, we show that satisfying first-order optimality is enough to show global optimality under condition **C1** for data independent W.

**Remark 4.3** *Due to Lemma 4.1 and corollary 4.2 then, rank of collection $h(u^i)u^{i^T}$ is invariant under any rotation.*

**Remark 4.4** *As a result of corollary above one can see that the collection of vectors $h(Wx^i)$ is full rank under the assumption that $W$ is non-singular, $x^i \in \mathbb{R}^d$ are independently sampled from Lebesgue measure and $h$ satisfies condition **C1**.*

**Remark 4.5** *Since collection $h(Wu^i)$ is also full rank, we can say that $z^i := h(W_1u^i)$ are independent and sampled from a Lebesgue measure for a non-singular matrix $W_1$. Applying the Lemma to $z^i$, we have collection of matrices $g(W_2z^i)z^{i^T}$ are full rank with measure 1 for non-singular $W_2$ and $g$ satisfying condition **C1**. So we see that for multiple hidden layers satisfying non-singularity, we can apply full rank property for collection of gradients with respect to outermost hidden layer.*

**Remark 4.6** *If $W \in \mathbb{R}^{n \times d}$ is such that $n \leq d$ and $W$ is full row rank, then we can extend its basis to create $W'$ and apply corollary 4.2 to get that $h(W' u^i) u^{i^T}$ is full rank with measure 1. So this implies that $h(W u^i) u^{i^T}$ must have been full rank with probability 1 otherwise we will have contradiction.*

**Remark 4.7** *We can extend corollary 4.2 to a general result that $h(W u^i) u^{i^T}$ has rank $\min\{rank(W)d, N\}$ with measure 1 by removing dependent rows and using remark 4.6.*

## 5 CONVERGENCE RESULTS FOR STOCHASTIC GRADIENT DESCENT

Even though we have proved that collection $\left\{ h(W u^i) u^{i^T} \right\}_{i=1}^{N}$ is full rank in the previous section, we need such $W$ to be independent of data. In general, any algorithm will use data to find $W$ and it appears that results in previous section are not meaningful in practice.

However, the analysis of Lemma 4.1 motivates the idea that stochastic noise of $\theta$ might help in obtaining the required properties of $W$ and $D$ along the trajectory of Algorithm 1. In this section we first prove that by using random noise in stochastic gradient on $\theta$ gives a non-singular $W_k$ in every iteration. Then using this fact, we prove that matrix $D$ generated along the algorithm is also full rank. The proof techniques are very similar to proof of Lemma 4.1.

Later on, we will also show that overall algorithm will converge to approximate first-order optimal solution to Problem (2.1) by using smoothness properties. It should be noted however that this can not guarantee convergence to a global optimal solution. To prove such a result, one needs to analyze the smallest singular value of random matrix $D$, defined in (3.3). More specifically, we have to show that $\sigma_{\min}(D)$ decreases at the rate slower than the first-order convergence rate of the algorithm so that the overall algorithm converges to the global optimal solution. Even if it is very difficult to prove such a result in theory, we think that such an assumption about $\sigma_{\min}(D)$ is reasonable in practice. Now we analyze the algorithm. For the sake of simplicity of notation, let us define

$$\xi^{[k]} := \{\xi^1_{[N_i]}, \ldots, \xi^k_{[N_i]}\} \tag{5.1}$$

and

$$\xi^j_{[N_i]} = \{\xi^j_1 \ \cdots \ \xi^j_{N_i}\}, \tag{5.2}$$

where $N_i$ is the inner iteration count in Algorithm 1. Essentially $\xi^{[k]}$ contains the record of all random samples used until the $k$-th outer iteration in Algorithm 1 and $\xi^j_{[N_i]}$ contains record of all random samples used in the inner iterations of $j$-th outer iteration.

**Lemma 5.1** $\Pr\{\exists \ \mathbf{v} \text{ such that } W_k \mathbf{v} = \mathbf{0} \big| \xi^{[k-1]}\} = 0, \forall \ k \geq 0$, *where $W_k$ are matrices generated by Algorithm 1 and measure $\Pr\{.\big|\xi^{[k-1]}\}$ is w.r.t. random variables $\xi^k_{[N_i]}$.*

Now that we have proved that $W_k$'s generated by the algorithm are full rank, we show that matrix $D$ generated along the trajectory of the algorithm is full rank for any finite number of iterations. We use techniques inspired from Lemma 4.1 but this time we use Lebesgue measure over $\Theta$ rather than data. Over randomness of $\Theta$, we can show that our algorithm will not produce any $W$ such that corresponding matrix $D$ is rank deficient. Since $\Theta$ is essentially designed to be independent of data so we will not produce rank deficient $D$ throughout the process of randomized algorithm.

**Lemma 5.2** *Suppose $W = W' + D_v Z$ where $D_v := diag(v[i], i \in [d])$ and $v$ is a random vector with Lebesgue measure in $\mathbb{R}^d$. $W', Z \in \mathbb{R}^{d \times d}$ and $Z \neq \mathbf{0}$. Let $h$ be a function which follows condition **C1**. Also assume that $W$ is full rank with measure 1 over randomness of $v$. Then $h(W u^i) u^{i^T}$ is full rank with measure 1 over randomness of $v$.*

**Lemma 5.3** *Collection of matrices $h'(W_{k+1} u^i) u^{i^T}$ are full rank with measure 1, where the measure is over randomness of $\xi^{k+1}_{[N_i]}$*

**Proof.** We know that

$$W_{k+1} = W_k + \gamma_k \Theta_{k+1} \sum_{j=1}^{N} h'(W_k u^j) u^{j^T} (v^i - \theta^T h(W_k u^j)).$$

Now apply Lemma 5.2 to obtain the required result. □

Note that Lemma 5.3 is very similar to the result in section 4. Some remarks are in order.

**Remark 5.4** *As a result of Lemma 5.3 above, one can see that collection of vectors $h(W_k u^i)$ is full rank for all finite iterations of Algorithm 1.*

**Remark 5.5** *If we have a neural network with multiple hidden layer, we can assume that inner layers are initialized to arbitrary full rank matrices and we are optimizing w.r.t. outermost hidden layer. Corollary 4.2 and Remark 4.4 give us that input to outermost hidden layer are independent vectors sampled from some lebesgue measure. Then applying Algorithm 1 to optimize w.r.t. outermost hidden layer will give us similar results as mentioned in Lemma 5.3.*

Hence we showed that algorithm will generate full rank matrix $D$ for any finite iteration.
Now to prove convergence of the algorithm, we need to analyze the function $f$ (defined in (2.1)) itself. We show that $f$ is a Lipschitz smooth function for any given instance of data $\{u^i, v^i\}_{i=1}^N$. This will give us a handle to estimate convergence rates for the given algorithm.

**Lemma 5.6** *Assuming that $h : \mathbb{R} \to \mathbb{R}$ is such that its gradients, hessian as well as values are bounded and data $\{u^i, v^i\}_{i=1}^N$ is given then there exists a constant $L$ such that*

$$\left\|\nabla_W f(W_1, \theta) - \nabla_W f(W_2, \theta)\right\|_F \le L \left\|W_1 - W_2\right\|_F. \tag{5.3}$$

*Moreover, a possible upper bound on $L$ can be as follows:*

$$L \le \frac{1}{N} \theta_{\max} \Big( L_{h'} \big( \sum_{i=1}^N \|u^i\|_2^2 |v^i| \big) + \sqrt{2d} L_{hh'} \|\theta\|_2 \big( \sum_{i=1}^N \|u^i\|_2^2 \big) \Big)$$

**Remark 5.7** *Before staing the proof, we should stress that assumptions on $h$ is satisfied by most activation functions e.g., sigmoid, sigmoid symmetric, gaussian, gaussian symmetric, elliot, elliot symmetric, tanh, Erf.*

**Remark 5.8** *Note that one can easily estimate value of $L$ given data and $\theta$. Moreover, if we put constraints on $\left\|\theta\right\|_2$ then $L$ is constant in every iteration of the algorithm 1. As mentioned in section 3, this will provide an easier way to analyze the algorithm.*

**Lemma 5.9** *Assuming that scalar function $h$ is such that $|h(\cdot)| \le u$ then there exists $L_\theta$ s.t.*

$$\left\|\nabla_w f(W, \theta_1) - \nabla_w f(W, \theta_2)\right\|_2 \le L_\theta \left\|\theta_1 - \theta_2\right\|_2 \tag{5.4}$$

Notice that Lemma 5.9 gives us value of $L_\theta$ irrespective of value of $W$ or data. Also observe that $f(W, \cdot)$ is convex function since hessian

$$\nabla_\theta^2 f(W, \theta) = \frac{1}{N} \sum_{i=1}^N h(W u^i) h(W u^i)^T,$$

which is the sum of positive semidefinite matrices. By Lemma 5.9, we know that $f(W, \cdot)$ is smooth as well. So we can use following convergence result provided by Lan (2012) for stochastic composite optimization. A simplified proof can be found in appendix.

**Theorem 5.10** *Assume that stepsizes $\beta_i$ satisfy $0 < \beta_i \le 1/2L_\theta, \forall\, i \ge 1$. Let $\{\theta_{i+1}^{av}\}_{i \ge 1}$ be the sequence computed according to Algorithm 1. Then we have,*

$$\mathbb{E}[f(W_k, \theta_{i+1}^{av}) - f(W_k, \theta_{W_k}^*)] \le K_0(i), \, \forall\, i \ge 1, \forall\, k \ge 0, \tag{5.5}$$

*where $K_0(i) := \left( \sum_{\tau=1}^i \beta_\tau \right)^{-1} \left[ \|\overline{\theta}_1 - \theta_{W_k}^*\|_2^2 + \sigma^2 \sum_{\tau=1}^i \beta_i^2 \right]$ where $\overline{\theta}_1$ is the starting point for inner iteration and $\sigma$ is defined in (3.5).*

Now we look at a possible strategy of selecting stepsize $\beta_i$. Suppose we adopt a constant stepsize policy then we have $\beta_i = \beta, \forall\, i \in [N_i]$. Then we have

$$\mathbb{E}[f(W_k, \theta_{N_i+1}^{av}) - f(W_k, \theta_{W_k}^*)] \leq \frac{\left\|\bar{\theta}_1 - \theta_{W_k}^*\right\|^2}{N_i\beta} + \sigma^2\beta.$$

Now if we choose

$$\beta = \min\left\{\frac{1}{2L_\theta}, \sqrt{\frac{1}{N_i\sigma^2}}\right\}, \tag{5.6}$$

we get

$$\mathbb{E}[f(W_k, \theta_{N_i+1}^{av}) - f(W_k, \theta_{W_k}^*)] \leq \left\|\bar{\theta}_1 - \theta_{W_k}^*\right\|^2\left[\frac{2L_\theta}{N_i} + \frac{\sigma}{\sqrt{N_i}}\right] + \frac{\sigma}{\sqrt{N_i}}.$$

By Lemma 5.6, the objective function for neural networks is Lipschitz-smooth with respect to the hidden layer, i.e., it satisfies eq (5.3). Notice that it is equivalent to saying

$$\left|f(W_2, w) - f(W_1, w) - \langle\nabla_W f(W_1, w), W_2 - W_1\rangle\right| \leq \frac{L}{2}\left\|W_1 - W_2\right\|_F^2, \quad \forall\, W_1, W_2 \in \mathbb{R}^{d\times d}. \tag{5.7}$$

Since we have a handle on the smoothness of objective function, we can provide a convergence result for the overall algorithm.

**Theorem 5.11** *Suppose $\gamma_k < \frac{2}{L}$ then we have*

$$\mathbb{E}\left[\min_{k=0,\ldots,N}\left\|\nabla f_W(W_k, \theta_{k+1})\right\|_F^2\right] \leq \frac{f(W_0, \theta_0) + \sum\limits_{k=0}^{N_o}\left(\sum\limits_{\tau=1}^{N_i}\beta_\tau^k\right)^{-1}\left[\frac{R^2}{2} + \sum\limits_{\tau=1}^{N_i}\frac{\beta_\tau^{k^2}\sigma^2}{2(1-L_\theta\beta_\tau^k)}\right]}{\sum\limits_{k=0}^{N_o}(\gamma_k - L/2\gamma_k^2)},$$

$$\tag{5.8}$$

*where $R/2$ is the radius of origin centred ball, $\mathcal{R}$ in algorithm, defined as $\mathcal{R} := \{r \in \mathbb{R}^d : \|r\|_2 \leq \frac{R}{2}\}$.*

In view of Theorem 5.11, we can derive a possible way of choosing $\gamma_k, \sigma$ and $N_i$ to obtain a convergence result. More specifically, if $N_i = N_o, \sigma = \frac{1}{\sqrt{N_i}}, \gamma_k = \frac{1}{L}$ and $\beta_\tau^k$ is chosen according to (5.6) then we have

$$\mathbb{E}\left[\min_{k=0,\ldots,N}\left\|\nabla_W f(W_k, \theta_{k+1})\right\|_F^2\right] \leq \frac{2L\left(f(W_0, \theta_0) + R^2(L_\theta + 1/2) + 1\right)}{N_o}$$

Note that in the algorithm 1, we have proved that having a stochastic noise helps keeping matrix $D$ full rank for all finite iterations. Then in Theorem 5.11, we showed a methodical way of achieving approximate first order optimal. So essentially at the end of the finitiely many steps of algorithm 1, we have a point $W$ which satisfies full rank property of $D$ and is approximately first order optimal. We think this kind of result can be extended to variety of different first order methods developed for Lipschitz-smooth non-convex optimization problems. More specifically, accelerated gradient method such as unified accelerated method proposed by Ghadimi et al. (2015) or accelerated gradient method by Ghadimi & Lan (2016) can be applied in outer iteration. We can also use stochastic gradient descent method for outer iteration. For this, we need a stochastic algorithm that is designed for non-convex and Lipschitz smooth function optimization. Randomized stochastic gradient method, proposed by Ghadimi & Lan (2013), Stochastic variance reduction gradient method (SVRG) by Reddi et al. or Simplified SVRG by Allen-Zhu & Hazan can be employed in outer iteration. Convergence of these new algorithms will follow immediately from the convergence results of respective studies. Some work may be needed to prove that they hold matrix $D$ full rank. We leave that for the future work. We also leave the problem of proving a bound on singular value for future. This will close the gap between empirical results and theory.

Value of Lipschitz constant, $L$, puts a significant impact on the running time of the algorithm. Notice that if $L$ increases then correspondingly $N_o$ and $N_i$ increase linearly with $L$. So we need methods by which we can reduce the value of the estimate of $L$. One possible idea would be to use $l_1$-ball for feasible region of $\theta$. More specifically, if $\mathcal{R} = \mathcal{B}^1(R/2)$ then we can possibly enforce sparsity on $\theta$ which will allow us to put better bound on $L$.

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

# A    PROOFS OF AUXILIARY RESULTS

In this appendix, we provide proofs for auxiliary results.

## A.1    PROOF OF LEMMA 4.1

The result is trivially true for d =1, we will show this using induction on d.

Define $\mathbf{v}^i := \text{vect}(h(x^i)x^{i^T}), i \in [N]$. Note that it suffices to prove independence of vector $\mathbf{v}^i, i \in [N]$ for $N \leq d^2$.

Now for sake of contradiction assume that $\mathbf{v}^i, i \in [N]$, are linearly dependent with positive joint measure on $x^i, i \in [N]$ which is equivalent to positive measure on individual $x^i, \forall\, i \in [N]$ due to independence of vectors $x^i$.

Since $x^i$'s are sampled from Lebesgue measure so positive measure on $x^i, \forall i \in [N]$, implies there exists a $d$-dimensional volume for each $x^i$ such that corresponding $\mathbf{v}^i$ are linearly dependent. We can assume volume to be $d$-dimensional hyper-cuboid $Z^i := \{x \in \mathbb{R}^d : a^i < x < b^i\}, \forall\, i \in [N]$ (otherwise we can inscribe a hyper-cuboid in that volume). Notice that since $Z^i$ is a d-dimensional hyper-cuboid so $a^i[k] < b^i[k], \forall i \in [N], \forall k \in [d]$. Moreover, for any collection satisfying $x^i \in Z^i$, corresponding collection of vector $\mathbf{v}^i$ are linearly dependent, i.e.,

$$\mathbf{v}^1 = \mu_2\mathbf{v}^2 + \cdots + \mu_N\mathbf{v}^N, \quad \text{such that } \forall i \in [N], x^i \in Z^i. \tag{A.1}$$

Noticing the definition of $Z^1$, we can choose $\epsilon > 0$ s.t. $\widetilde{x}^1 := x^1 + \epsilon e_1 \in Z^1$. Since we ensure that $\widetilde{x}^1 \in Z^1$ then by (A.1) we have

$$\widetilde{v}^1 := \text{vect}(h(\widetilde{x}^1)\widetilde{x}^{1^T}) = \mu_2'\mathbf{v}^2 + \cdots + \mu_N'\mathbf{v}^N. \tag{A.2}$$

So using (A.1) and (A.2) we get

$$\widetilde{v}^1 - \mathbf{v}^1 = \lambda_2\mathbf{v}^2 + \cdots + \lambda_N\mathbf{v}^N. \tag{A.3}$$

Since $h(x^i)x^{i^T}[j,k] = h(x^i[j])x^i[k]$, we have $h(x^1)x^{1^T}[j,k] = h(\widetilde{x}^1)\widetilde{x}^{1^T}[j,k], \forall j \in \{2,\ldots,d\}, k \in \{2,\ldots,d\}$. So we have $(d-1)^2$ components of $\widetilde{v}^1 - \mathbf{v}^1$ are zero. Let us define:

$$w^1 = \begin{bmatrix} (x^1[1] + \epsilon)h(x^1[1] + \epsilon) - x^1[1]h(x^1[1]) \\ \epsilon h(x^1[2]) \\ \vdots \\ \epsilon h(x^1[d]) \\ x^1[2](h(x^1[1] + \epsilon) - h(x^1[1])) \\ \vdots \\ x^1[d](h(x^1[1] + \epsilon) - h(x^1[1])) \end{bmatrix} \left.\rule{0pt}{3.5em}\right\}(2d-1) \quad, \quad z^1 = \begin{bmatrix} 0 \\ \vdots \\ 0 \end{bmatrix} \left.\rule{0pt}{2em}\right\}(d-1)^2 \quad,$$

and notice that $\widetilde{v}^1 - \mathbf{v}^1 = \begin{bmatrix} w^1 \\ z^1 \end{bmatrix}$. Since $\epsilon > 0, w^1 \neq \mathbf{0}$ with measure 1.

Let $y^i := x^i[2:d]$ then last $(d-1)^2$ equations in (A.3) gives us

$$\lambda_2 h(y^2){y^2}^T + \cdots + \lambda_N h(y^N){y^N}^T = z^1 = \mathbf{0} \tag{A.4}$$

By definition we have $y^i \in \mathbb{R}^{d-1}$ are independently sampled from $(d-1)$-dimensional Lebesgue measure. So by inductive hypothesis, rank of collection of matrices $h(y^i){y^i}^T, i \in \{2,\ldots,N\} = \min\{(d-1)^2, N-1\}$. So if $N-1 \leq (d-1)^2$ then $\lambda_2 = \cdots = \lambda_N = 0$ with measure 1, then by (A.3) we have $w^1 = \mathbf{0}$ with measure 1, which is contradiction to the fact that $w^1 \neq \mathbf{0}$ with measure 1. This gives us

$$N > (d-1)^2 + 1 \tag{A.5}$$

Notice that (A.4) in its matrix form can be written as linear system

$$\begin{bmatrix} \text{vect}(h(y^2){y^2}^T) & \ldots & \text{vect}(h(y^N){y^N}^T) \end{bmatrix} \begin{bmatrix} \lambda_2 \\ \vdots \\ \lambda_N \end{bmatrix} = \mathbf{0} \tag{A.6}$$

By (A.6), we have that vector of $\lambda$'s lies in the null space of the matrix. Finally by inductive hypothesis and (A.5) we conclude that the dimension of that space is $N - 1 - (d-1)^2 \{> 0\}$. Let $\mathbf{u}^1, \ldots, \mathbf{u}^{N-1-(d-1)^2} \in \mathbb{R}^{N-1}$ be the basis of that null space i.e.

$$\begin{bmatrix} \text{vect}(h(y^2){y^2}^T) & \ldots & \text{vect}(h(y^N){y^N}^T) \end{bmatrix} \mathbf{u}^j = 0, \quad \forall j \in \{1, N-1-(d-1)^2\}$$

Define $t^i \in \mathbb{R}^{2d-1}$ as:

$$t^i := \begin{bmatrix} x^i[1]h(x^i[1]) \\ \vdots \\ x^i[1]h(x^i[d]) \\ x^i[2]h(x^i[1]) \\ \vdots \\ x^i[d]h(x^i[1]) \end{bmatrix}$$

then we can rewrite (A.3) as

$$\begin{bmatrix} w^1 \\ z^1 \end{bmatrix} = \begin{bmatrix} t^2 & \ldots & t^N \\ \text{vect}(h(y^2){y^2}^T) & \ldots & \text{vect}(h(y^N){y^N}^T) \end{bmatrix} \begin{bmatrix} \mathbf{u}^1 & \ldots & \mathbf{u}^{N-1-(d-1)^2} \end{bmatrix} \begin{bmatrix} \widehat{\lambda}_2 \\ \vdots \\ \widehat{\lambda}_{N-(d-1)^2} \end{bmatrix} \tag{A.7}$$

which implies that

$$w^1 = \widehat{\lambda}_2 \widehat{v}^2 + \cdots + \widehat{\lambda}_{N-(d-1)^2} \widehat{v}^{N-(d-1)^2} \tag{A.8}$$

where $\widehat{v}^i = \begin{bmatrix} t^2 & \ldots & t^N \end{bmatrix} \mathbf{u}^{i-1}, \ i = 2, \ldots, N-(d-1)^2$ and $z^1$ part of the equation is already satisfied due to selection of null space.

Since $N \leq d^2 \Rightarrow N - 1 - (d-1)^2 \leq 2d-2$ then $2d-1$ equations specified in (A.8) are consistent in $\leq (2d-2)$ variables. Hence we get linearly dependent equations $\forall x_1^1 \in (a_1^1, b_1^1)$ and $\epsilon$ small enough. Since $x^2, \ldots, x^N$ are kept constant, $\mathbf{v}^2, \ldots, \mathbf{v}^N$ are constant. So $t^2, \ldots, t^N$ are constants and we can choose the same basis of null space $\mathbf{u}^1, \ldots, \mathbf{u}^{N-1-(d-1)^2}$. Hence we have $\widehat{v}^2, \ldots, \widehat{v}^{(N-(d-1)^2)}$ are constant. Let us define the set $\mathcal{S}$ to be the index set of linearly independent rows of matrix $\begin{bmatrix} \widehat{v}^2 & \ldots & \widehat{v}^{N-(d-1)^2} \end{bmatrix}$ and every other row is a linear combination of rows in $\mathcal{S}$. Since (A.8) is consistent so the same combination must be valid for the rows of $w^1$.

Now if $N \leq d^2 - 1$ then number of variables in (A.8) is $\leq 2d-3$ but number of equations is $2d-1$, therefore at least two equations are linearly dependent on other equation. This implies last $(2d-2)$ equations then function must be dependent on each other:

$$\epsilon \sum_{j=2}^{d} \alpha_j h(x^{(1)}[j]) + \left( h(x^{(1)}[1] + \epsilon) - h(x^{(1)}[1]) \right) \sum_{j=2}^{d} \beta_j x^{(1)}[j] = 0$$

for some fixed combination $\alpha_j, \beta_j$. If we divide above equation by $\epsilon$ and take the limit as $\epsilon \to 0$ then we see that $h$ satisfies following differential equation on interval $(a_1^1, b_1^1)$:

$$h'(x) = c_1$$

which is a contradiction to the condition **C1**!

Clearly this leaves only one case i.e. $N = d^2$ and $(2d - 1)$ equations must satisfy dependency of the following form for all $x_1^{(1)} \in (a_1^{(1)}, b_1^{(1)})$:

$$(x^{(1)}[1] + \epsilon)h(x^{(1)}[1] + \epsilon) - x^{(1)}[1]h(x^{(1)}[1])$$
$$= \epsilon \sum_{j=2}^{d} \alpha_j h(x^{(1)}[j]) + \left( h(x^{(1)}[1] + \epsilon) - h(x^{(1)}[1]) \right) \sum_{j=2}^{d} \beta_j x^{(1)}[j]$$

Again by similar arguments, the combination is fixed. Let $H(x) = xh(x)$ then dividing above equation by $\epsilon$ and taking the limit as $\epsilon \to 0$, we can see that $h$ satisfies following differential equation:

$$H'(x) = c_1 + c_2 h'(x) \Rightarrow (x - c_2)h'(x) + h(x) = c_1 \tag{A.9}$$

which is again a contradiction to the condition **C1**

So we conclude that for $N \le d^2$ there does not exist hyper-cuboids $Z^i$ such that $\mathrm{vol}(Z^i) > 0$ and for all $x^i \in Z^i$, corresponding $\mathbf{v}^i$ are linearly dependent. So we get rank of collection $\{\mathbf{v}^i\}_{i=1}^{N} = \min\{N, d^2\}$ with measure 1.

## A.2 Proof of corollary 4.2

Let us define $x := Wu$ be another random variable. Since $W$ is full rank and $u$ has Lebesgue measure $\Rightarrow x$ has Lebesgue measure.

Now we claim that the collection $h(Wu^i)u^{i^T}$ is full rank iff the collection $h(x^i)x^{i^T}$ is full rank. This can observed as follows:

$$\sum_{i=1}^{N} \lambda_i h(x^i)x^{i^T} = 0 \Leftrightarrow \left\{ \sum_{i=1}^{N} \lambda_i h(Wu^i)u^{i^T} \right\} W^T = 0$$
$$\Leftrightarrow \sum_{i=1}^{N} \lambda_i h(Wu^i)u^{i^T} = 0$$

Here the second statement follows from the fact $W$ is a non-singular matrix.

Now by Lemma 4.1 we have that collection $h(x^i)x^{i^T}$ is linearly independent with measure 1. So $h(Wu^i)u^{i^T}$ is linearly independent with measure 1.

Since any rotation is $U$ is a full rank matrix so we have the result.

## A.3 Proof of Lemma 5.1

This is true for $k = 0$ trivially since we are randomly sampling matrix $W_0$. We now show this by induction on $k$.

Recall that gradient of $f(W, \theta)$ with respect to $W$ can be written as $\sum_{i=1}^{N} \{v^i - \theta^T h(Wu^i)\}\left(h'(Wu^i) \odot \theta\right)u^{i^T}$. Notice that in effect, we are multiplying $i$-th row of the rank one matrix $h'(Wu^i)u^{i^T}$ by $i$-th element of vector $\theta$. So this can be rewritten as a matrix product

$$\sum_{i=1}^{N} \{v^i - \theta^T h(Wu^i)\}\Theta h'(Wu^i)u^{i^T},$$

where $\Theta := \mathrm{diag}\{\theta[i], i = 1, \ldots, d\}$. So iterative update of the algorithm can be given as

$$W_{k+1} = W_k - \gamma_k \Theta_{k+1} \nabla_W f(W_k, \theta_{k+1}), \quad \forall k \ge 0.$$

Notice that given $\xi^{[k]}$, vector $\theta_{k+1}$ and corresponding diagonal matrix $\Theta_{k+1}$ are found by SGD in the inner loop so $\theta_{k+1}$ is a random vector. More specifically, since $\{\xi_i^{k+1}\}_{i=1}^{N_i}$ is sequence of

independent $d$-dimensional isotropic Gaussian vectors. Hence the distribution of $\xi^{k+1} = \{\xi_i^{k+1}\}_{i=1}^{N_i}$ induces a Lebesgue measure on random variable $\{\theta_{k+1} | \xi^{[k]}\}$

Given $\xi^{[k]}$ then $W_k$ is deterministic quantity.

For the sake of contradiction, take any vector $\mathbf{v}$ that is supposed to be in the null space of $W_{k+1}$ with positive probability.

$$W_{k+1} = W_k - \gamma_k \nabla_W f(W_k, \theta_{k+1})$$

$$= W_k - \gamma_k \sum_{i=1}^{N} \Theta_{k+1}(v^i - \theta_{k+1}^T h(W_k u^i)) h'(W_k u^i) u^{i^T}.$$

$$\Rightarrow W_{k+1}\mathbf{v} = W_k\mathbf{v} - \gamma_k \sum_{i=1}^{N} \Theta_{k+1}(v^i - \theta_{k+1}^T h(Wu^i)) h'(W_k u^i) u^{i^T}\mathbf{v} = 0.$$

$$\Rightarrow W_k\mathbf{v} = \Theta_{k+1}\sum_{i=1}^{N}(\lambda_i v^i - r_i^T \theta_{k+1}) h'(W_k u^i) \qquad \text{setting } \lambda_i = \gamma_k(\mathbf{v}^T u^i), r_i = \lambda_i h(W_k u^i)$$

$$= \Theta_{k+1}\left[ \sum_{i=1}^{N} \lambda_i v^i h'(W_k u^i) - \left(\sum_{i=1}^{N} h'(W_k u^i) r_i^T\right)\theta_{k+1} \right].$$

Now the last equation is of the form

$$b = \Theta_{k+1}[w - M\theta_{k+1}], \tag{A.10}$$

where $b = W_k\mathbf{v}$, $w = \sum_{i=1}^{N} \lambda_i v^i h'(W_k u^i)$, $M = \sum_{i=1}^{N} h'(W_k u^i) r_i^T$.

Suppose we can find such $\theta$ with positive probability. Then we can find hypercuboid $Z := \{x \in \mathbb{R}^d | a < x < b\}$ such that any $\theta_{k+1}$ in given hypercuboid can solve equation (A.10). By induction we have $b \neq \mathbf{0}$. We may assume $b[1] \neq 0$. Then to get contradiction on existence of $Z$, we observe that first equation in (A.10) is:

$$b[1] = \theta_{k+1}[1]\left(w[1] - \sum_{j=2}^{d} M[1,j]\theta_{k+1}[j]\right) - M[1,1]\theta_{k+1}[1]^2, \quad \forall \theta_{k+1} \in (a,b). \tag{A.11}$$

Hence if we fix $\theta_{k+1}[i] \in (a[i], b[i]), i = 2, \ldots, d$ then (A.11) holds for all $\theta_{k+1}[1] \in (a[1], b[1])$. So we conclude that $b[1] = w[1] + \sum_{j=2}^{d} M[1,j]\theta_{k+1}[j] = M[1,1] = 0$. But $b[1]$ can not be 0. Hence we arrive at a contradiction to the assumption that there existed a hypercuboid $Z$ containing solutions of (A.10).

Since measure on $\theta_{k+1}$ was induced by $\{\xi_i^{k+1}\}_{i=1}^{N_i}$ so we conclude that $\Pr\{\exists \, \mathbf{v} \text{ such that } W_{k+1}\mathbf{v} = 0 | \xi^{[k]}\} = 0, \forall \, k \geq 0$.

## A.4 PROOF OF LEMMA 5.2

We use induction on $d$. For $d = 1$ this is trivially true. Now assume this is true for $d - 1$. We will show this for $d$.

Let $z^i := Wu^i = W'u^i + D_v Zu^i$. For simplicity of notation define $t^i := Zu^i$. Due to simple linear algebraic fact provided by full rank property of $W$ we have rank of collection $(h(Wu^i)u^{i^T} = $ rank of collection $h(z^i)z^{i^T}$. For the sake of contradiction, say the collection is rank deficient with positive probability then there exists $d$-dimensional volume $\mathcal{V}$ such that for all $v \in \mathcal{V}$, we have $h(Wu^i)u^{i^T}$ is not full rank where $W := W(v) = W' + D_v Z$. Without loss of generality, we may assume $d$-dimensional volume to be a hypercuboid $\mathcal{V} := \{x \in \mathbb{R}^d | a < x < b\}$ (if not then we can inscribe a hypercuboid in that volume). Let us take $v \in \mathcal{V}$ and $\varepsilon$ small enough such that $\widehat{v} := v + \varepsilon e_1 \in \mathcal{V}$. Correspondingly we have $z^i$ and $\widehat{z}^i$. Note that $\widehat{z}^i = z^i + \varepsilon t^i[1]$. So in essence, a small $\varepsilon$ change in $v[1]$ causes $\varepsilon t^i[1]$ change in vector $z^i[1]$.

Let $\mathbf{v}^i = \text{vect}(h(z^i)z^{i^T})$. Similarly, $\widehat{\mathbf{v}}^i = \text{vect}(h(\widehat{z}^i)\widehat{z}^{i^T})$. So we can divide $\mathbf{v}^i = \begin{bmatrix} c^i \\ g^i \end{bmatrix}$ such that $c^i \in \mathbb{R}^{2d-1}$ and $g^i \in \mathbb{R}^{(d-1)^2}$. Here

$$
c^i := \begin{bmatrix} h(z^i[1])z^i[1] \\ h(z^i[2])z^i[1] \\ \vdots \\ h(z^i[d])z^i[1] \\ h(z^i[1])z^i[2] \\ \vdots \\ h(z^i[1])z^i[d] \end{bmatrix}, \quad g^i := \text{vect}(h(y^i)y^{i^T}), \quad y^i := z^i[2:d]
$$

Similarly we also have $\widehat{\mathbf{v}}^i = \begin{bmatrix} \widehat{c}^i \\ \widehat{g}^i \end{bmatrix}$. Now by the act that $v, \widehat{v}$ corresponding to $z, \widehat{z}$ are in $\mathcal{V}$, and our assumption of linear dependence for all $v \in \mathcal{V}$ we get

$$
\mathbf{v}^1 = \mu_2 \mathbf{v}^2 + \cdots + \mu_N \mathbf{v}^N \tag{A.12}
$$

$$
\widehat{\mathbf{v}}^1 = \widehat{\mu}_2 \widehat{\mathbf{v}}^2 + \cdots + \widehat{\mu}_N \widehat{\mathbf{v}}^N \tag{A.13}
$$

Now notice that $y^i = \widehat{y}^i, \forall i \in [N]$. So $g^i = \widehat{g}^i, \forall i \in [N]$. Also by induction on $d-1$, we have that the rank of collection $g^2, \ldots, g^N \geq (d-1)^2$. So we can rewrite matrix $[g^2 \ldots g^N] := [G \quad \widetilde{G}]$ such that $G \in \mathbb{R}^{(d-1)^2 \times (d-1)^2}$ is an invertible matrix and rewrite one part of equation (A.12) as $g^1 = [G \quad \widetilde{G}] \begin{bmatrix} \widetilde{\mu} \\ \mu \end{bmatrix}$. Hence we can replace $\widetilde{\mu} = G^{-1}(g^1 - \widetilde{G}\mu) = G^{-1}g^1 - G^{-1}\widetilde{G}\mu$. Essentially the vector $\begin{bmatrix} \widetilde{\mu} \\ \mu \end{bmatrix}$ is completely defined by parameter $\mu \in \mathbb{R}^{N-1-(d-1)^2}$. Similarly we have $\widehat{\widetilde{\mu}} = G^{-1}g^1 - \widetilde{G}\widehat{\mu}$, so vector $\begin{bmatrix} \widehat{\widetilde{\mu}} \\ \widehat{\mu} \end{bmatrix}$ is completely defined by $\widehat{\mu} \in \mathbb{R}^{N-1-(d-1)^2}$. So essentially we have satisfied one part of equations (A.12) and (A.13). Notice that since we are moving only one coordinate of random vector $v$ i.e. $v[1] \in (a[1], b[1])$ (by $\varepsilon$ incremental changes) keeping all other elements of $v$ constant so we will have $y^i$ as constants which implies $g^i, G, \widetilde{G}$ are constant. So for the sake of simplicity of notation we define $l := G^{-1}g^1 \in \mathbb{R}^{(d-1)^2}$ and $R := G^{-1}\widetilde{G} \in \mathbb{R}^{(d-1)^2 \times (N-1-(d-1)^2)}$

Now, we look at the remaining part of two equation (A.12),(A.13):

$$
c^1 = \mu_2 c^2 + \cdots + \mu_N c^N,
$$

$$
\widehat{c}^1 = \widehat{\mu}_2 \widehat{c}^2 + \cdots + \widehat{\mu}_N \widehat{c}^N,
$$

which can be rewritten as

$$
c^1 = [C \quad \widetilde{C}] \begin{bmatrix} l - R\mu \\ \mu \end{bmatrix} = Cl - CR\mu + \widetilde{C}\mu, \tag{A.14}
$$

$$
\widehat{c}^1 = [\widehat{C} \quad \widehat{\widetilde{C}}] \begin{bmatrix} l - R\widehat{\mu} \\ \widehat{\mu} \end{bmatrix} = \widehat{C}l - \widehat{C}R\widehat{\mu} + \widehat{\widetilde{C}}\widehat{\mu}. \tag{A.15}
$$

After (A.15) − (A.14), we have

$$
(\widehat{C} - C)l - (\widehat{C} - C)R\mu - \widehat{C}R(\widehat{\mu} - \mu) + (\widehat{\widetilde{C}} - \widetilde{C})\mu + \widehat{\widetilde{C}}(\widehat{\mu} - \mu) = \widehat{c}^1 - c^1. \tag{A.16}
$$

Now note that (A.16), characterizes incremental changes in $C, \widetilde{C}, \mu$ due to $\varepsilon$. So taking the limit as $\varepsilon \to 0$, we have

$$
c^{1'} = C'l - C'R\mu - CR\mu' + \widetilde{C}'\mu + \widetilde{C}\mu'.
$$

$$
[c^{1'} \quad C'] \begin{bmatrix} 1 \\ -l \end{bmatrix} = (-CR + \widetilde{C})\mu' + (-C'R + \widetilde{C}')\mu.
$$

$$
\Rightarrow [c^1 \quad C] \begin{bmatrix} 1 \\ -l \end{bmatrix} = (-CR + \widetilde{C})\mu. \tag{A.17}
$$

Here, last equation is due to product rule in calculus. In (A.17), we see that we have $2d-1$ equations and $N-1-(d-1)^2$ unknowns at every point. If $N \leq d^2$ then $N-1-(d-1)^2 \leq 2d-2$. So at least one equation should depend on others. But as we have shown earlier, $h$ satisfying condition **C1** does not have row dependence. So we arrive at the required contradiction for $N \leq d^2$. That completes the proof.

### A.5    PROOF OF LEMMA 5.6

Assume that all the gradients in this proof are w.r.t. $W$ then we know that

$$-\nabla f(W, \theta)[j, k] = \frac{1}{N} \sum_{i=1}^{N} \{v^i - \theta^T h(Wu^i)\} h'(W[j, :]u^i)\theta[j]u^i[k]$$

Notice that $\|W\|_F = \|\text{vect}(W)\|_2$. Also notice that if $W = ab^T$ then $\|W\|_F = \|a\|_2.\|b\|_2$
Let us define vector $a^i$ s.t. $a^i[j] = \theta[j]h'(W[j, :]u^i)(v^i - \theta^T h(Wu^i))$ so we have

$$-(\nabla f(W_1) - \nabla f(W_2))_{jk} = \frac{1}{N} \sum_{i=1}^{N} u_k^i(a_1^i[j] - a_2^i[j])$$

$$\Rightarrow -(\nabla f(W_1) - \nabla f(W_2)) = \frac{1}{N} \sum_{i=1}^{N} (a_1^i - a_2^i)u^{i^T}$$

$$\Rightarrow \|\nabla f(W_1) - \nabla f(W_2)\|_F \leq \frac{1}{N} \sum_{i=1}^{N} \|u^i\|_2.\|a_1^i - a_2^i\|_2, \tag{A.18}$$

where the last inequality follows from Cauchy-Schwarz inequality.
So if we can show Lipschitz constant $L_i$ on $\|a_1^i - a_2^i\|_2, \forall i$ then we are done.
Let $\theta_{\max} := \max_{j} |\theta_j|$, then

$$\left|(a_1^i - a_2^i)[j]\right| = |\theta_j|.\left|h'(W_1[j, :]u^i)(v^i - \theta^T h(W_1 u^i)) - h'(W_2[j, :]u^i)(v^i - \theta^T h(W_2 u^i))\right|$$

$$\leq \theta_{\max}\left|h'(W_1[j, :]u^i)(v^i - \theta^T h(W_1 u^i)) - h'(W_2[j, :]u^i)(v^i - \theta^T h(W_2 u^i))\right|$$

$$\Rightarrow \|a_1^i - a_2^i\|_2 \leq \theta_{\max}\left\|v^i\left(h'(W_1 u^i) - h'(W_2 u^i)\right) - \left(h(W_1 u^i)h'(W_1 u^i)^T - h(W_2 u^i)h'(W_2 u^i)^T\right)\theta\right\|_2$$

$$\leq \theta_{\max}\left\{\left\|v^i\left(h'(W_1 u^i) - h'(W_2 u^i)\right)\right\|_2\right.$$

$$\left. + \left\|(h'(W_1 u^i)h(W_1 u^i)^T - h'(W_2 u^i)h(W_2 u^i)^T)\theta\right\|_2\right\}.$$

Suppose the Lipschitz constants for the first and second term are $L_{i,L}$ and $L_{i,R}$ respectively. Then $L_i = \theta_{\max}(L_{i,L} + L_{i,R})$ and possible upper bound on value of $L$ would become $\frac{1}{N} \sum_{i=1}^{N} \|u^i\|_2 L_i$.
We now analyse existence of $L_{i,L}$
Since the Hessian of scalar function $h(\cdot)$ is bounded so we have $h'(x)$ is Lipschitz continuous with constant $L_{h'}$. Let $r_1, r_2$ be two row vectors then we claim $\|h'(r_1 x) - h'(r_2 x)\|_2 \leq L_{h'}\|x\|_2.\|r_1 - r_2\|_2, \forall r_1, r_2$ because:

$$\|h'(r_1 x) - h'(r_2 x)\|_2 \leq L_{h'}|r_1 x - r_2 x| \leq L_{h'}\|x\|_2\|r_1 - r_2\|_2$$

From the relation above we have the following:

$$\left\|h'(W_1 u^i) - h'(W_2 u^i)\right\|_2^2 = \sum_{j=1}^{d} \left(h'(W_1[j, :]u^i) - h'(W_2[j, :]u^i)\right)^2$$

$$\leq L_{h'}^2\|u^i\|_2^2 \sum_{j=1}^{d} \|W_1[j, :] - W_2[j, :]\|_2^2 = L_{h'}^2\|u^i\|_2^2\|W_1 - W_2\|_F^2$$

$$\Rightarrow L_{i,L} = L_{h'}\|u^i\|_2|v^i|. \tag{A.19}$$

Now we focus our attention to second term. Notice the simple fact that

$$\|W_1 - W_2\|_2 \le \|W_1 - W_2\|_F = \|\text{vect}(W_1 - W_2)\|_2. \tag{A.20}$$

Define $\mathbf{v} := W_1 u^i, \mathbf{u} := W_2 u^i$, then we have

$$\|\mathbf{v} - \mathbf{u}\|_2 = \left\|(W_1 - W_2)u^i\right\|_2 \le \left\|W_1 - W_2\right\|_2 \cdot \left\|u^i\right\|_2 \le \left\|u^i\right\|_2 \cdot \left\|\text{vect}(W_1 - W_2)\right\|_2, \tag{A.21}$$

and

$$\left\|\left(h'(W_1 u^i)h(W_1 u^i)^T - h'(W_2 u^i)h(W_2 u^i)^T\right)\theta\right\|_2$$
$$\le \left\|\theta\right\|_2 \cdot \left\|h'(W_1 u^i)h(W_1 u^i)^T - h'(W_2 u^i)h(W_2 u^i)^T\right\|_2$$
$$= \left\|\theta\right\|_2 \cdot \left\|h'(\mathbf{v})h(\mathbf{v})^T - h'(\mathbf{u})h(\mathbf{u})^T\right\|_2$$
$$\le \left\|\theta\right\|_2 \cdot \left\|h'(\mathbf{v})h(\mathbf{v})^T - h'(\mathbf{u})h(\mathbf{u})^T\right\|_F.$$

The latter inequality implies that

$$\left\|(h'(W_1 u^i)h(W_1 u^i)^T - h'(W_2 u^i)h(W_2 u^i)^T)\theta\right\|_2^2$$
$$\le \|\theta\|_2^2 \left(\sum_{i,j=1}^d h'(\mathbf{v}[i])h(\mathbf{v}[j]) - h'(\mathbf{u}[i])h(\mathbf{u}[j])\right)^2.$$

Now let us define a 2-D function $H(x_1, x_2) = h(x_1)h'(x_2)$. Then $\nabla H(x_1, x_2) = \begin{bmatrix} h'(x_1)h'(x_2) \\ h(x_1)h''(x_2) \end{bmatrix}$
so under given assumptions, $\|\nabla H(\cdot)\|_2$ is bounded. Let that bound be $L_{hh'}$.
Now by mean value theorem, we have

$$H(x_1, x_2) - H(y_1, y_2) = \nabla H(\xi)^T \{(x_1, x_2) - (y_1, y_2)\}$$
$$\Rightarrow \left|H(x_1, x_2) - H(y_1, y_2)\right|^2 \le \left\|\nabla H(\xi)\right\|_2^2 \cdot \left\{(x_1 - y_1)^2 + (x_2 - y_2)^2\right\}$$
$$\le L_{hh'}^2 \left\{(x_1 - y_1)^2 + (x_2 - y_2)^2\right\}$$
$$\text{So } \left\|\left\{h'(W_1 u^i)h(W_1 u^i)^T - h'(W_2 u^i)h(W_2 u^i)^T\right\}\theta\right\|_2^2$$
$$\le \|\theta\|_2^2 \left(\sum_{i,j=1}^d h'(\mathbf{v}[i])h(\mathbf{v}[j]) - h'(\mathbf{u}[i])h(\mathbf{u}[j])\right)^2$$
$$\le \|\theta\|_2^2 \sum_{i,j=1}^d L_{hh'}^2 \left((\mathbf{v}[i] - \mathbf{u}[i])^2 + (\mathbf{v}[j] - \mathbf{u}[j])^2\right)$$
$$= 2d L_{hh'}^2 \|\theta\|_2^2 \|\mathbf{v} - \mathbf{u}\|_2^2 \tag{A.22}$$

It then follows from (A.20),(A.21) and (A.22) that

$$\left\|(h'(W_1 u^i)h(W_1 u^i)^T - h'(W_2 u^i)h(W_2 u^i)^T)\theta\right\|_2$$
$$\le \sqrt{2d}L_{hh'}\|\theta\|_2 \cdot \|u^i\|_2 \cdot \|W_1 - W_2\|_F$$

So you get that $L_{i,R} = \sqrt{2d}L_{hh'}\|\theta\|_2\|u^i\|_2$
Finally, using (A.18), (A.19) and (A.22), we get a possible finite upper bound on the value of $L$:

$$L \le \frac{1}{N}\theta_{\max}\left(L_{h'}\left(\sum_{i=1}^N \|u^i\|_2^2 |v^i|\right) + \sqrt{2d}L_{hh'}\|\theta\|_2\left(\sum_{i=1}^N \|u^i\|_2^2\right)\right)$$

Also note that this bound is valid even if $W$ is not a square matrix.

### A.6 PROOF OF LEMMA 5.9

Noting that

$$-\nabla_\theta f(W,\theta) = \frac{1}{N} \sum_{i=1}^{N} \{v^i - \theta^T h(Wu^i)\} h(Wu^i),$$

we have

$$\left\| \nabla_\theta f(W,\theta_1) - \nabla_w f(W,\theta_2) \right\|_2$$

$$= \left\| \frac{1}{N} \sum_{i=1}^{N} \left[ \{v^i - \theta_1^T h(Wu^i)\} h(Wu^i) - \{v^i - \theta_2^T h(Wu^i)\} h(Wu^i) \right] \right\|_2$$

$$= \left\| \frac{1}{N} \sum_{i=1}^{N} \left[ \{-h(Wu^i)h(Wu^i)^T \theta_1 + h(Wu^i)h(Wu^i)^T \theta_2\} \right] \right\|_2$$

$$= \left\| \frac{1}{N} \sum_{i=1}^{N} h(Wu^i)h(Wu^i)^T (\theta_2 - \theta_1) \right\|_2$$

$$\leq \left\| \frac{1}{N} \sum_{i=1}^{N} h(Wu^i)h(Wu^i)^T \right\|_2 . \left\| \theta_1 - \theta_2 \right\|_2$$

$$= \frac{1}{N} \left\| \sum_{i=1}^{N} h(Wu^i)h(Wu^i)^T \right\|_2 . \left\| \theta_1 - \theta_2 \right\|_2$$

$$= \frac{1}{N} \lambda_{\max} \left( \sum_{i=1}^{N} h(Wu^i)h(Wu^i)^T \right) . \left\| \theta_1 - \theta_2 \right\|_2$$

$$\leq \frac{1}{N} \left\{ \sum_{i=1}^{N} \lambda_{\max} \left( h(Wu^i)h(Wu^i)^T \right) \right\} . \left\| \theta_1 - \theta_2 \right\|_2 \qquad \because \text{Weyl's Inequality}$$

$$= \frac{1}{N} \left\{ \sum_{i=1}^{N} \left\| h(Wu^i) \right\|_2^2 \right\} . \left\| \theta_1 - \theta_2 \right\|_2$$

$$\leq u^2 d \left\| \theta_1 - \theta_2 \right\|_2$$

where $u_1$ and $u_2$ are upper bounds on scalar functions $|h(\cdot)|$ and $|h'(\cdot)|$ respectively and $d$ is row-dimension of $W$.

### A.7 PROOF OF THEOREM 5.11

We know by Lemma 5.6 that $f(\cdot, \theta)$ is a Lipschitz smooth function. So using (5.7) we have

$$f(W_{k+1}, \theta_{k+1}) \leq f(W_k, \theta_{k+1}) + \frac{L}{2} \left\| \text{vect}(W_{k+1} - W_k) \right\|^2$$

$$+ \left\langle \text{vect}(\nabla_W f(W_k, \theta_{k+1})), \text{vect}(W_{1_{k+1}} - W_{1_k}) \right\rangle$$

$$= f(W_k, \theta_{k+1}) - \left( \gamma_k - \frac{L}{2} \gamma_k^2 \right) \left\| \text{vect}(\nabla_W f(W_k, \theta_{k+1})) \right\|^2$$

$$\leq f(W_k, \theta_k) + \left( \sum_{\tau=1}^{N_i} \beta_\tau^k \right)^{-1} \left[ \frac{1}{2} \left\| \theta_k - \theta_{W_k}^* \right\|_2^2 + \sum_{\tau=1}^{N_i} \beta_\tau^k \left\langle \xi_\tau^k, \theta_{W_k}^* - \overline{\theta}_\tau^k \right\rangle \right.$$

$$\left. + \sum_{\tau=1}^{N_i} \frac{\beta_\tau^{k^2} \left\| \xi_\tau^k \right\|^2}{2(1 - L_\theta \beta_\tau^k)} \right] - \left( \gamma_k - \frac{L}{2} \gamma_k^2 \right) \left\| \text{vect}(\nabla_W G(W_k, \theta_{k+1})) \right\|^2, \qquad \text{(A.23)}$$

where the last inequality follows from equation (A.28) and (A.29).

From (3.6), we have $\|\theta\| \leq R/2$ so $L$ is constant for each outer iteration. Summing (A.23) from

$k = 0$ to $N_o$ and dividing both side by $\sum_{k=0}^{N_o}(\gamma_k - \frac{L}{2}\gamma_k^2)$, we get

$$\min_{k=0,\ldots,N}\left\|\nabla_W f(W_k,\theta_{k+1})\right\|_F^2 \le \sum_{k=0}^{N_o}\frac{(\gamma_k - \frac{L}{2}\gamma_k^2)\left\|\mathrm{vect}(\nabla_{W_1}f(W_k,\theta_{k+1}))\right\|^2}{\sum_{k=0}^{N_o}(\gamma_k - \frac{L}{2}\gamma_k^2)}$$

$$\le \frac{f(W_0,\theta_0) + \sum_{k=0}^{N_o}\left(\sum_{\tau=1}^{N_i}\beta_\tau^k\right)^{-1}\left[\frac{R^2}{2} + \sum_{\tau=1}^{N_i}\left\{\beta_\tau^k\left\langle \xi_\tau^k, \theta_{W_k}^* - \overline{\theta}_\tau^k\right\rangle + \frac{\beta_\tau^{k2}\left\|\xi_\tau^k\right\|^2}{2(1-L_\theta\beta_\tau^k)}\right\}\right]}{\sum_{k=0}^{N_o}(\gamma_k - L/2\gamma_k^2)}.$$

Now taking expectation with respect to $\xi^{[N_o]}$ (which is defined in (5.1)), we have

$$\mathbb{E}\left[\left\langle \xi_\tau^k, \theta_{W_k}^* - \overline{\theta}_\tau^k\right\rangle\Big|\xi^{[k-1]}\cup\xi_{[\tau-1]}^k\right] = 0,$$

which implies $\mathbb{E}_{\xi^{[N_o]}}\left[\left\langle \xi_\tau^k, \theta_{W_k}^* - \overline{\theta}_\tau^k\right\rangle\right] = 0$. We also have $\mathbb{E}_{\xi^{[N_o]}}\left[\left\|\xi_\tau^k\right\|^2\right] \le \sigma^2$, and hence

$$\mathbb{E}\left[\min_{k=0,\ldots,N}\left\|\nabla_W f(W_k,\theta_{k+1})\right\|_F^2\right] \le \frac{f(W_0,\theta_0) + \sum_{k=0}^{N_o}\left(\sum_{\tau=1}^{N_i}\beta_\tau^k\right)^{-1}\left[\frac{R^2}{2} + \sum_{\tau=1}^{N_i}\frac{\beta_\tau^{k2}\sigma^2}{2(1-L_\theta\beta_\tau^k)}\right]}{\sum_{k=0}^{N_o}(\gamma_k - L/2\gamma_k^2)}.$$

## A.8 PROOF OF THEOREM 5.10

For sake of simplicity of notation, we define $\mathbf{f}(\cdot) := f(W_k,\cdot), \mathbf{g}(\cdot) := \nabla\mathbf{f}(\cdot) = \nabla_\theta f(W_k,\cdot)$ and $G_{W_k}(\overline{\theta}_\tau,\xi_\tau^k) := \mathbf{G}_\tau$. Then from (3.4) and Algorithm 1 we get

$$\mathbf{G}_\tau = \mathbf{g}(\overline{\theta}_\tau) + \xi_\tau^k. \tag{A.24}$$

Also define $d_\tau := \overline{\theta}_{\tau+1} - \overline{\theta}_\tau$.

Notice that $\overline{\theta}_{\tau+1}$ is optimal solution to the problem

$$\min_{u\in\mathbb{R}^d}\beta_\tau\left\langle \mathbf{G}_\tau, u - \overline{\theta}_\tau\right\rangle + \frac{1}{2}\left\|u - \overline{\theta}_\tau\right\|_2^2, \tag{A.25}$$

by simply writing first order necessary condition for problem (A.25). Also we note that objective function in (A.25) is strongly convex with parameter 1. Then we have

$$\beta_\tau\left\langle \mathbf{G}_\tau, d_\tau\right\rangle + \frac{1}{2}\left\|d_\tau\right\|_2^2 + \frac{1}{2}\left\|u - \overline{\theta}_{\tau+1}\right\|_2^2 \le \beta_\tau\left\langle \mathbf{G}_\tau, u - \overline{\theta}_\tau\right\rangle + \frac{1}{2}\left\|u - \overline{\theta}_\tau\right\|_2^2. \tag{A.26}$$

We will use eq (A.26) along with smoothness and convexity of the function $\mathbf{f}$ to get the final convergence result. Notice that due to smoothness, we have

$$\beta_\tau\mathbf{f}(\overline{\theta}_{\tau+1}) \le \beta_\tau[\mathbf{f}(\overline{\theta}_\tau) + \left\langle \mathbf{g}(\overline{\theta}_\tau), d_\tau\right\rangle + \frac{L_\theta}{2}\|d_\tau\|^2]$$

$$= \beta_\tau[\mathbf{f}(\overline{\theta}_\tau) + \left\langle \mathbf{g}(\overline{\theta}_\tau), d_\tau\right\rangle] + \frac{1}{2}\|d_\tau\|^2 - \frac{(1-L_\theta\beta_\tau)}{2}\|d_\tau\|^2.$$

then due to (A.24) we have,

$$\beta_\tau\mathbf{f}(\overline{\theta}_{\tau+1}) \le \beta_\tau[\mathbf{f}(\overline{\theta}_\tau) + \left\langle \mathbf{G}_\tau, d_\tau\right\rangle] - \beta_\tau\left\langle \xi_\tau^k, d_\tau\right\rangle + \frac{1}{2}\|d_\tau\|^2 - \frac{(1-L_\theta\beta_\tau)}{2}\|d_\tau\|^2$$

$$\le \beta_\tau[\mathbf{f}(\overline{\theta}_\tau) + \left\langle \mathbf{G}_\tau, d_\tau\right\rangle] + \frac{1}{2}\|d_\tau\|^2 - \frac{(1-L_\theta\beta_\tau)}{2}\|d_\tau\|^2 + \beta_\tau\|\xi_\tau^k\|.\|d_\tau\|$$

$$\le \beta_\tau\mathbf{f}(\overline{\theta}_\tau) + \left[\beta_\tau\left\langle \mathbf{G}_\tau, d_\tau\right\rangle + \frac{1}{2}\|d_\tau\|^2\right] + \frac{\beta_\tau^2\|\xi_\tau^k\|^2}{2(1-L_\theta\beta_\tau)}.$$

By (A.26) we have

$$
\begin{aligned}
\beta_\tau \mathbf{f}(\overline{\theta}_{\tau+1}) &\leq \beta_\tau \mathbf{f}(\overline{\theta}_\tau) + \beta_\tau \left\langle \mathbf{G}_\tau, u - \overline{\theta}_\tau \right\rangle + \frac{1}{2}\|u - \overline{\theta}_\tau\|^2 - \frac{1}{2}\|u - \overline{\theta}_{\tau+1}\|^2 + \frac{\beta_\tau^2 \|\xi_\tau^k\|^2}{2(1 - L_\theta \beta_\tau)} \\
&= \left[ \beta_\tau \mathbf{f}(\overline{\theta}_\tau) + \beta_\tau \left\langle \mathbf{g}(\overline{\theta}_\tau), u - \overline{\theta}_\tau \right\rangle \right] + \beta_\tau \left\langle \xi_\tau^k, u - \overline{\theta}_\tau \right\rangle \\
&\quad + \frac{1}{2}\|u - \overline{\theta}_\tau\|^2 - \frac{1}{2}\|u - \overline{\theta}_{\tau+1}\|^2 + \frac{\|\beta_\tau^2 \xi_\tau^k\|^2}{2(1 - L_\theta \beta_\tau)} \\
&\leq \beta_\tau \mathbf{f}(u) + \beta_\tau \left\langle \xi_\tau^k, u - \overline{\theta}_\tau \right\rangle + \frac{1}{2}\|u - \overline{\theta}_\tau\|^2 - \frac{1}{2}\|u - \overline{\theta}_{\tau+1}\|^2 + \frac{\beta_\tau^2 \|\xi_\tau^k\|^2}{2(1 - L_\theta \beta_\tau)}.
\end{aligned}
\tag{A.27}
$$

Last equation is due to convexity of function $\mathbf{f}$. So using (A.27) we have

$$
\sum_{\tau=1}^{i} \beta_\tau \left[ \mathbf{f}(\overline{\theta}_{\tau+1}) - \mathbf{f}(\theta_{W_k}^*) \right] \leq \frac{1}{2}\|\overline{\theta}_1 - \theta_{W_k}^*\|^2 + \sum_{\tau=1}^{i} \left[ \beta_\tau \left\langle \xi_\tau^k, \theta_{W_k}^* - \overline{\theta}_\tau \right\rangle + \frac{\beta_\tau^2 \|\xi_\tau^k\|^2}{2(1 - L_\theta \beta_\tau)} \right]. \tag{A.28}
$$

Note that from convexity of $\mathbf{f}$, we get

$$
\mathbf{f}(\theta_{i+1}^{av}) - \mathbf{f}(\theta_{W_k}^*) \leq \left( \sum_{\tau=1}^{i} \beta_\tau \right)^{-1} \sum_{\tau=1}^{i} \left[ \beta_\tau \left[ \mathbf{f}(\overline{\theta}_{\tau+1}) - \mathbf{f}(\theta_{W_k}^*) \right] \right]. \tag{A.29}
$$

Moreover, noting the definition of $\xi_{[\tau]}^k$ in (5.2) so we have,

$$
\mathbb{E}\left[ \left\langle \xi_\tau^k, \theta_{W_k}^* - \overline{\theta}_\tau \right\rangle \big| \xi_{[\tau-1]}^k \right] = 0, \tag{A.30}
$$

and from (3.5) we get $\mathbb{E}\left[\|\xi_\tau^k\|^2\right] \leq \sigma^2$. Hence using this relation and noting $1 - L_\theta \beta_\tau \geq \frac{1}{2}$, (A.28), (A.29) and (A.30) we prove the result.

