# OpenReview forum: "Theoretical properties of the global optimizer of two-layer Neural Network"
_ICLR.cc/2018/Conference — Reject_

### Official Review · AnonReviewer2 · 2017-11-26
**This paper aims to study some of the theoretical properties of the global optima of single-hidden layer neural networks and also the convergence to such a solution. I think there are some interesting arguments made in the paper. However, as I started reading beyond intro I increasingly got the sense that this paper is somewhat incomplete e.g. while certain claims are made (abstract/intro) the theoretical justification are rather far from these claims.**

**Rating:** 7
**Confidence:** 5

**Review:**

This paper aims to study some of the theoretical properties of the global optima of single-hidden layer neural networks and also the convergence to such a solution. I think there are some interesting arguments made in the paper e.g. Lemmas 4.1, 5.1, 5.2, and 5.3. However, as I started reading beyond intro I increasingly got the sense that this paper is somewhat incomplete e.g. while certain claims are made (abstract/intro) the theoretical justification are rather far from these claims. Of course there is a chance that I might be misunderstanding some things and happy to adjust my score based on the discussions here.

Detailed comments:
1) My main concern is that the abstract and intro claims things that are never proven (or even stated) in the rest of the paper
Example 1 from abstract:
“We show that for a wide class of differentiable activation functions (this class involved “almost” all functions which are not piecewise linear), we have that first-order optimal solutions satisfy global optimality provided the hidden layer is non-singular.”

This is certainly not proven and in fact not formally stated anywhere in the paper. Closest result to this is Lemma 4.1 however, because the optimal solution is data dependent this lemma can not be used to conclude this.

Example 2 from intro when comparing with other results on page 2:
The authors essentially state that they have less restrictive assumptions in the form of the network or assumptions on the data (e.g. do not require Gaussianity). However as explained above the final conclusions are also significantly weaker than this prior literature so it’s a bit of apples vs oranges comparison.

2) Page 2 minor typos
We study training problem -->we study the training problem
In the regime training objective--> in the regime the training objective

3) the basic idea argument and derivative calculations in section 3 is identical to section 4 of Soltan...et al

4) Lemma 4.1 is nice, well done! That being said it does not seem easy to make it (1) quantifiable (2) apply to all W. It would also be nice to compare with Soudry et. al.

5) Argument on top of page 6 is incorrect as the global optima is data dependent and hence lemma 4.1 (which is for a fixed matrix) does not apply

6) Section 5 on page 6. Again the stated conclusion here that the iterates do not lead to singular W is much weaker than the claims made early on.

7) I haven’t had time yet to verify correctness of Lemmas 5.1, 5.2, and Lemma 5.3 in detail but if this holds is a neat argument to side step invertibility w.r.t. W, Nicely done!

8) What is the difference between Lemma 5.4 and Lemma 6.12 of Soltan...et al

9) Theorem 5.9. Given that the arguments in this paper do not show asymptotic convergence to a point where gradient vanishes and W is invertible why is the proposed algorithm better than a simple approach in which gradient descent is applied but a small amount of independent Gaussian noise is injected in every iteration over W. By adjusting the noise variance across time one can ensure a result of the kind in Theorem 5.9 (Of course in the absence of a quantifiable version of Lemma 4.1 which can apply to all W that result will also suffer from the same issues).

---

> ### Author Response · Authors · 2017-12-15
> **Paper Revision**
>
> We have added a revision to the paper. After reading comments of all reviewers, there was a realization that the way paper was written may have caused some confusion. We tried to address those concerns. Please go through it. New explanations are written in red. We will greatly appreciate your feedback. Specific replies to points you raised:
> 1) My main concern is that the abstract and intro claims things that are never proven (or even stated) in the rest of the paper
> Example 1 from abstract: “We show that for a wide class of differentiable activation functions (this class involved “almost” all functions which are not piecewise linear), we have that first-order optimal solutions satisfy global optimality provided the hidden layer is non-singular.” This is certainly not proven and in fact not formally stated anywhere in the paper. Closest result to this is Lemma 4.1 however, because the optimal solution is data dependent this lemma can not be used to conclude this.
> Example 2 from intro when comparing with other results on page 2: The authors essentially state that they have less restrictive assumptions in the form of the network or assumptions on the data (e.g. do not require Gaussianity). However as explained above the final conclusions are also significantly weaker than this prior literature so it’s a bit of apples vs oranges comparison.
> Response:
> We appreciate the Reviewer’s question. We now try to clarify these issues in the revised version of the paper.
> In the abstract, we have added the word arbitrary first order optimal points to signify that in this case we are looking at data-independent W. Then in the immediate next sentence we make it clear that we extend these results to $W_k$'s found along a trajectory of an algorithm thereby alleviating some of the concerns. Then in section 5, once we state Lemma 5.3, we have added remarks 5.4 and 5.5 addressing the issue of implementing this algorithmic framework for multiple hidden layer. Note that we use some of the results we proved for arbitrary $W$ while stating remark 5.5.
> We have also added comparison to previous results in the Introduction and Section 3. We make sure that we have apples vs oranges part of the results stated in the explanation.
>
> 2) Page 2 minor typos
> Response:
> Above mentioned typos are fixed in new version.
>
> 3) the basic idea argument and derivative calculations in section 3 is identical to section 4 of Soltan...et al
> Response:
> We are not sure about the basic idea because they also look second order conditions. But the derivative is indeed same.
>
> 4) Lemma 4.1 is nice, well done! That being said it does not seem easy to make it (1) quantifiable (2) apply to all W. It would also be nice to compare with Soudry et. al.
> Response:
> Thanks for the comments. We have added comparison to Soudry et al. in section 3. Also a motivation to look at arbitrary W is added at the beginning of section 4. This will relieve some concerns of readers.
>
> 5) Argument on top of page 6 is incorrect as the global optima is data dependent and hence lemma 4.1
> Response:
> This is a similar concern to the previous ones. We have added more materials in Section 4 and Section 5 to alleviate these concerns.
>
> 7) I haven’t had time yet to verify correctness of Lemmas 5.1, 5.2, and Lemma 5.3 in detail but if this holds is a neat argument to side step invertibility w.r.t. W, Nicely done!
> Response:
> Thanks. Also please read the new remarks added at the end of these results. This is to extend this result for training outermost hidden layer of multilayer neural network. We use some results in section 4 for arbitrary W to prove/state these remarks.
>
> (Reply continued in next comment)

---

> > ### Author Response · Authors · 2017-12-15
> > **Revision to the paper**
> >
> > 8) What is the difference between Lemma 5.4 and Lemma 6.12 of Soltan...et al
> > Response:
> > We now add some discussions about the difference in the introduction. We also believe you meant Lemma 6.14 of arxiv version of that paper.
> >
> > 9) Theorem 5.9. Given that the arguments in this paper do not show asymptotic convergence to a point where gradient vanishes and W is invertible why is the proposed algorithm better than a simple approach in which gradient descent is applied but a small amount of independent Gaussian noise is injected in every iteration over W. By adjusting the noise variance across time one can ensure a result of the kind in Theorem 5.9
> > Response:
> > This is now Theorem 5.11. We agree that injecting Gaussian noise to W may obtain similar result, but this approach will slightly modify the problem to be solved. On the other other hand, we develop an algorithmic framework which alternates between the gradient descent step with respect to W and the SGD procedure for the output layer \theta. A few SGD steps will help to guarantee the non-singularity of W, but would not modify the original problem since we make sure the original function value is descent. The overall scheme is also inspired to the iterative layer-by-layer training that has been used in practice. We concluded the results talking about the pluses and minuses of this algorithm and possibility of extending this to multiple others.
> > In the end, we think to prove global optimality especially for the highly nonlinear activation functions studied in this paper is challenging problem to solve in totality without making any additional assumptions. But at the same time our work should inspire new ideas for tackling this challenge.

---

> > > ### Comment · AnonReviewer2 · 2018-01-12
> > > **Reply**
> > >
> > > Thanks for your response and clarification. I think the authors have clarified that their result is assuming a data independent W at each iteration as I had thought.  I think this needs to be stated more clearly in the abstract and intro. I was actually confused by "arbitrary" in the abstract and the sentence " hence even if the hidden layer variables are data dependent, we still get required properties" in the intro. In fact I got the impression that they meant the opposite i.e. their results hold even when W does depend on data. I recommend using "data-independent" instead of arbitrary and rephrasing the latter sentence.
> > >
> > > While I believe this paper leaves much to be desired I am increasing my rating to 7 as I believe this paper is a step forward and in the right direction and has interesting ideas and insights. My overall recommendation is acceptance after addressing the concerns above.

---

### Official Review · AnonReviewer3 · 2017-12-02

**Rating:** 4
**Confidence:** 5

**Review:**

The paper studies the theoretical properties of the two-layer neural networks.

To summarize the result, let's use the theta to denote the layer closer to the label, and W to denote the layer closer to the data.

The paper shows that
a) if W is fixed, then with respect to the randomness of the data, with prob. 1, the Jacobian matrix of the model is full rank
b) suppose that we run an algorithm with fresh samples, then with respect to the randomness of the k-th sample, we have that with prob. 1, W_k is full rank, and the Jacobian of the model is full rank.

It's know (essentially from the proof of Carmon and Soudry) that if the Jacobian of the model is full rank for any matrix W w.r.t the randomness of the data, then all stationary points are global. But the paper cannot establish such a result.

The paper is not very clear, and after figuring out what it's doing, I don't feel it really provides many new things beyond C-S and Xie et al.

The paper argues that it works for activation beyond relu but result a) is much much weaker than the one with for all quantifier for W. result b) is very sensitive to the exactness of the events (such as W is exactly full rank) --- the events that the paper talks just naturally never happen as long as the density of the random variables doesn't degenerate.

As the author admitted, the results don't provide any formal guarantees for the convergence to a global minimum. It's also a bit hard for me to find the techniques here provide new ideas that would potentially lead to resolving this question.

--------------------

additional review after seeing the author's response:

The author's response pointed out some of the limitation of Soudry and Carmon, and Xie et al's which I agree. However, none of this limitation is addressed by this paper (or addressed in a misleading way to some extent.)  The key technical limitation is the dependency of the local minima on the weight parameters. Soudry and Carmon addresses this in a partial way by using the random dropout, which is a super cool idea. Xie et al couldn't address this globally but show that the Jacobian is well conditioned for a class of weights. The paper here doesn't have either and only shows that for a single fixed weight matrix, the Jacobian is well-conditioned.

I don't also see the value of extension to other activation function. To some extent this is not consistent with the empirical observation that relu is very important for deep learning.

Regarding the effect of randomness, since the paper only shows the convergence to a first-order optimal solution, I don't see why randomness is necessary. Gradient descent can converge to a first order optimal solution. (Indeed I have a typo in my previous review regarding "w.r.t. k-th sample", which should be "w.r.t. k-th update". ) Moreover, to justify the effect of the randomness, the paper should have empirical experiments.

I think the writing of the paper is also misleading in several places.

---

> ### Author Response · Authors · 2017-12-07
> **Response**
>
> The reviewer stated in part b) of the summary that we show "Jacobian matrix D is full rank w.r.t. k-th sample". We do not show this. We show that assuming data is given then w.r.t. randomness of stochastic gradient of theta, all the Jacobian matrices will be full rank. So this is essentially property of the algorithm and not of the data-sample (You can choose whatever distribution you want for the stochastic gradient of theta; whereas distribution of data is not something you can choose in practice). There is no new sample generated in every iteration of the algorithm. All the data is sampled in the beginning and used as constant throughout the algorithm. Also the events are systematically shown to be happening with probability 1.
> Soudry et al. paper talks about the Jacobian but they do not give an algorithmic characterization for which this Jacobian property will hold. On the other hand, we give a simple characterization and show that stochastic gradient algorithm achieves these characterizations. In fact, this algorithm gives robustness w.r.t. dependence on data of the characterization. (This is actually observed in practice: A standard gradient descent does not necessarily converge to a good first-order stationary point but a noisy (stochastic) gradient descent mostly converges to "good” first-order points). Our theorem justifies how having noise helps in maintaining the robustness w.r.t dependence on data.
> Soudry et al. look at leaky ReLu with randomisation as "dropout". This model fails when there is no dropout as mentioned in their paper itself. Our whole analysis is for models whose activation is not ReLu which is of course a different (and bigger) class of activation functions. Moreover, we do not use random dropouts in our activations. We show our results without changing model but giving randomization to the algorithmic process of finding the W.
> So in essence we show that why having noisy gradients is important and also show that you can actually play with algorithm rather than the NN model to get these robustness guarantees. Moreover, our algorithms alternate between the output layer and the hidden layer, partly explaining the success of the layer-by-layer training. The bigger idea might be to say: "Random noise in one layer helps in keeping robustness w.r.t. data dependence on other layer."
> On the side, we also analyze Lipschitz smoothness properties of the optimization problem and show a positive result which only depends on data. Such result has great algorithmic value since these constants can be computed and employed in the algorithm. Also note that these lipschitz constants are not probabilistic. Using this, we establish convergence results to an approximate-first-order optimality point which satisfies our characterization. (If it were an exact-first order optimality point then we would be at global optimal). The core of the problem of proving global optimality is proving that lowest singular value does not converge to 0 faster that rate of convergence of the algorithm itself.
> In this regard, indeed Xie et al. give a positive result but there is a catch. Xie et al also look at the same Jacobian matrix but here again, they only consider ReLu units. There results are indeed strong in the sense that they show lower bounds on smallest singular value of Jacobian matrix but for that they need two facts. One is to restrict the activation to ReLu so that they can find spectrum of corresponding Kernel matrix, second is a bound on discrepancy of weights W. These conditions are strong in the sense that it is difficult to implement algorithm which will satisfy these conditions. On other hand, our condition is simple, workable for almost all activations (except ReLu), can be shown to be true in simple algorithmic set up but it does not get us an actual bound on lowest singular value of Jacobian matrix. We think both results have their own theoretical pluses and minuses.

---

> ### Author Response · Authors · 2017-12-15
> **Revision to the paper**
>
> After reading your and other reviewers' comments, there was a realization that the way paper was written has caused some confusion. We have tried to address those concerns in the revised paper. New explanations are put in red color. The issue you mentioned about order of quantifier is mentioned in the section 4 with the motivation to look at it even though there is obvious problem of data dependency. We partially address those in section 5.
> We also added comparison to Xie et al. and Soudry et al. More details of this are in section 3 where we make our approach clear and differentiate our approach with theirs.
> Please go through the revised paper and we will greatly appreciate your feedback.
> Achieving convergence to the global minimum is a challenging problem, especially if we do not make any additional assumptions. But we think that our results will inspire new ideas to tackle this challenge.

---

### Official Review · AnonReviewer1 · 2017-12-13
**Good paper - but important clarifications are missing in the text**

**Rating:** 7
**Confidence:** 4

**Review:**

I only got access to the paper after the review deadline; and did not have a chance to read it until now. Hence the lateness and brevity.

The paper tackles an important theoretical question; and it offers results that are complementary to existing results (e.g., Soudry et al). However, the paper does not properly relate their results, assumptions in the context of the existing literature. Much explanation is needed in the author reply in order to clear these questions.

The work should not be evaluated from a practical perspective as it is of a theoretical nature.

I agree with most of the criticism raised by other reviewers. However, I also believe the authors managed to clear essentially of the criticism in they reply. The paper lacks in clarity as currently written.

The results are interesting, but more explanation is needed for the main message to be conveyed more clearly. I suggest 7, but the paper has a potential to become 8 in my eyes in a future resubmission.

---

> ### Author Response · Authors · 2018-01-04
> **Revision to the Paper**
>
> Thank you very much for the positive comments. We also appreciate that you went through our replies to other reviewers and took a holistic view of the paper. We understand that first version of the paper had caused some confusion among reviewers. So we have added a new version in an effort to address their criticism. New comments are in red color in latest revision to the paper which was uploaded on 15th Dec 2017.
> We would greatly appreciate if you could go through it and give your feedback

---

### Comment · AnonReviewer3 · 2017-11-21
**order of quantifier**

What's the order of the quantifier in Corollary 4.2? It seems that the samples are sampled after W is fixed? Then the result doesn't seem to be useful, because the whole point of previous work such as SC, XLS is to prove the uniform result so that W can be chosen after samples are fixed.

---

> ### Author Response · Authors · 2017-11-21
> **Convergence results and Corollary 4.2**
>
> In corollary 4.2, matrix W is independent of data. That is indeed a problem we notice and address in section 5. Immediately after lemma 5.1, we discuss the exact problem you are referring to. We use the ideas developed in lemma 4.1/corollary 4.2 to show that even though you assume that data is fixed, using the randomness of stochastic theta, one can show that Algorithm 1 is robust and W_k generated by this algorithm achieves the same guarantees as that of W in corollary 4.2. Robustness is derived solely from lebesgue measure on theta.

---

> > ### Comment · AnonReviewer3 · 2017-11-22
> > **order of quantifier**
> >
> > But then this lemma 5.2,5.3 is sensitive in the sense that the resulting matrix W_k can be full rank but it may be very ill-conditioned (e.g., the least singular value can converge to zero very fast?)?  I guess this is why you couldn't prove that it converges to a global minimum?

---

> > > ### Author Response · Authors · 2017-11-22
> > > **Global convergence**
> > >
> > > Yes, for global convergence we need to prove that smallest singular value of matrix D does not decrease at the rate faster than 1/N (It may go to zero but the rate should not be faster than 1/N). We were unable to show this for the general setting where a simple noise is added to theta. Any suggestion in that direction are most welcome.

---

### Decision · Program_Chairs · 2018-01-29
**ICLR 2018 Conference Acceptance Decision**

**Decision:**

Reject

**Comment:**

Understanding the quality of the solutions found by gradient descent for optimizing deep nets is certainly an important area of research. The reviewers found several intermediate results to be interesting.  At the same time, the reviewers unanimously have pointed out various technical aspects of the paper that are unclear, particularly new contributions relative to recent prior work. As such, at this time, the paper is not ready for ICLR-2018 acceptance.